# Synthesis, Modification and Characterization of Antimicrobial Textile Surface Containing ZnO Nanoparticles

**DOI:** 10.3390/polym12061210

**Published:** 2020-05-26

**Authors:** L. Martinaga Pintarić, M. Somogi Škoc, V. Ljoljić Bilić, I. Pokrovac, I. Kosalec, I. Rezić

**Affiliations:** 1Faculty of Textile Technology, Department of Applied Chemistry, University of Zagreb, 10000 Zagreb, Croatia; lela.pintaric@ttf.hr (L.M.P.); maja.somogyi@ttf.hr (M.S.Š.); 2Faculty of Pharmacy and Biochemistry, University of Zagreb, 10000 Zagreb, Croatia; vljoljic@pharma.hr (V.L.B.); ivan.pokrovac.fbf@gmail.com (I.P.); ikosalec@pharma.hr (I.K.)

**Keywords:** nanoparticles, coating, antibacterial resistance, antimicrobial activity, ZnO, optimization

## Abstract

In this research, a textile surface was modified by the sol–gel methodology with a new antimicrobial coating containing nanoparticles active against bacteria resistant to antibiotics. The effect of ultrasonic irradiation power (40 to 90 kHz), the concentration of reagents (nanoparticles, precursor and acids) and time (15 to 72 min) were investigated in relation to the structure, morphology and antimicrobial activity of coatings with zinc oxide nanoparticles. The relationship between the sonocatalytic performance and structure of the resultant modification was established by using various techniques, such as FTIR spectroscopy (FTIR) and scanning electron microscopy with an EDX detector (SEM-EDX), thin-layer chromatography (TLC) and antimicrobial effects were determined on selected model microorganisms. The homogeneity of layers with ZnO nanoparticles on samples was increased by increasing the ultrasonic irradiation power and time. The ultrasonic irradiation unify did not only unify both the structure and the morphology of samples, it also prevented the agglomeration of the nanoparticles. Moreover, under optimal conditions, an antimicrobial coating with ZnO nanoparticles, active against bacterial species *S. aureus* and *E. coli* was efficiently prepared. Results of the *Time-kill* methodology revieled excellent results starting after 6 hours of exposal to antimicrobialy functionalized cellulose polymer.

## 1. Introduction

According to some predictions, by 2050, more people could die from the infections caused by antibiotic-resistant bacteria than from cancer. In Europe 25,000 deaths per year and costs over EUR 1.5 billion are associated with resistant microorganisms [1]. In particular, dangerous infections include bacteria such as methicillin-resistant *Staphylococcus aureus* (MRSA), known as a “*super-bacteria*”, which is increasingly difficult to cope with due to its resistance to a wide range of antibiotic-based penicillin drugs (β-lactam antibiotics such as penams and cephalosporins). *Staphylococcus aureus* is a member of the *Staphylococcaceae* family of Gram-positive bacteria spherical forms and is one of the most significant pathogens in the world. It is an infection with a frequency ranging from 20 to 50 cases per 100,000 inhabitants per year, with 10 to 30% of infections ending with a deadly outcome. This number is greater than the sum of the deaths caused by Acquired Immunodeficiency Syndrome (AIDS), tuberculosis and viral hepatitis combined [2].

Metal oxide nanoparticles provide excellent antimicrobial, water resistance and protective properties. Among them, ZnO nanoparticles are one of the most important antimicrobial reagents. Therefore, cellulose materials with ZnO nanoparticles coatings can be found in different protective materials. The extraordinary antimicrobial activity of ZnO nanoparticles, which receives significant global interest, is the result of very small diameters of nanoparticles, which are far below the range of the microorganisms’ dimensions. Recent investigations have shown that nanosized ZnO can interact with both bacterial surfaces and/or with the bacterial cells, after it enters through the membrane [3]. ZnO nanoparticles showed antimicrobial activity on different microorganisms, including Gram-positive and Gram-negative bacteria, as well as on spores that are resistant to high temperature and high pressure [4].

In contrast to their significant antimicrobial effects, nanoparticles made of ZnO were found to be non-toxic to human cells. Moreover, their biocompatibility with human cells was proven in the investigation of Padmavathy and Vijayaraghavan [5]. The safety of ZnO and its compatibility with human skin make it a suitable additive for textiles and surfaces that are exposed to the human body [6,7]. This makes them significant potential antimicrobial agents that need to be explored. Currently, such investigations are in progress. ZnO nanoparticles show activity over a wide spectrum of bacterial species, as reported by many researchers [2,4,5,8,9,10,11,12,13,14,15,16]. Padmavathy and Vijayaraghavan investigated the antibacterial activity of ZnO nanoparticles with various particle sizes and reported that with a decreasing particle size the antibacterial efficiency of ZnO nanoparticles increased [5]. In this research, Gram-negative bacteria seemed to be more resistant to ZnO nanoparticles than Gram-positive bacteria, and the antibacterial activity of ZnO nanoparticles increased with decreasing particle size and increasing powder concentration. Moreover, the antibacterial effect of ZnO nanoparticles was time dependent.

ZnO nanoparticles are not only effective as antimicrobial agents. In addition, they hold unique optical, chemical sensing, semiconducting, electric conductivity, and piezoelectric properties [17], as well as high catalytic and high photochemical activities. ZnO is attributed with high optical absorption in the UVA and UVB regions (315–400 nm and 280–315 nm, respectively), and is used as a UV protector in protective clothing, cosmetics and beneficial in antibacterial responses [18].

The current market of protective and medical antibacterial textile is currently limited due to the high cost of such materials. Therefore, innovative technologies for production of antibacterial materials are investigated. As for now, none of them are used in large quantities due to the complexity of developed methods and high price of the products. However, ultrasonic irradiation is an important step in applying antibacterial particles during modification of textile materials.

Antimicrobial modification of polymer materials by nanoparticles is enhanced in an ultrasonic field. Abramova et al. coated the textile using the sol–gel method for the synthesis of the titanium dioxide nanoparticles in combination with zinc oxide nanoparticles from titanyl sulfate and zinc nitrate hexahydrate with the goal to assess the method application feasibility to obtain antibacterial coatings on textile materials [19]. Their resulting textile samples reduced the number of microorganisms of *Escherichia coli* by more than 99.99% and achieved an antibacterial activity of more than 1.9 [19] Akhavan and Montazer simultaneously loaded titanium dioxide nanoparticles during their sonosynthesis onto a cotton fabric [20]. In their work, titanium tetra isopropoxide (TTIP) was used as the precursor and ultrasonic irradiation was utilized as a tool for the synthesis of TiO_2_ at low temperature with an anatase structure, by loading nanoparticles onto the cotton fabric. The results confirmed good self-cleaning and UV-protection properties, and that the sonochemical method had no negative influence on the cotton fabric structure. In contrast, ultrasound used in sonoimmobilization on textile materials like wool enhances homogenous distribution and results in excellent antibacterial/antifungal properties with low negative effects on human dermal fibroblasts [21].

In addition, the mechanical properties of nanocomposite films can be improved by ultrasonic irradiation after the addition of cellulose/silica to the antimicrobial polymer membrane structure [22]. Moreover, the addition of reducing, stabilizing and binding agents such as the natural biopolymer Tragacanth gum can benefit the in situ synthesis of zinc oxide nanoparticles on the cotton fabric. Ultrasonic irradiation leads to the clean and easy synthesis of zinc oxide nanoparticles over a short time and at a low temperature [23]. The finished cotton fabric has efficient antimicrobial protection against *Staphylococcus aureus, Escherichia coli* and *Candida albicans*, with inhibition zones of 3.3 ± 0.1, 3.1 ± 0.1 and 3.0 ± 0.1 mm [23]. The modification of the morphology of ZnO nanoparticles, for example, into thorn-like ZnO nanoparticles, shows even more promising antimicrobial results [24].

Nanomaterial toxicity effects depend on particle composition, size, and shape. The main mechanism of toxicity of nanoparticles is via oxidative species that damage proteins and DNA, as well as the catalytic properties, optical properties, and electrical conductivity of cells. During the investigation of the toxicity of nanoparticles, catalase, glutathione, and superoxide dismutase have been associated with the defense system to oxidative stress, according to Barata and his coworkers [25]. Liu et al. proved that nanoparticles of ZnO, CuO, TiO_2_, and Au are toxic to the daphnids of Daphnia magna and early life stages of zebrafish [26]. Mostly nanoparticles with diameters of 30 nm (ZnO) and 20 nm (TiO_2_) induced more toxicity than larger dimensions (50 nm ZnO and 30 nm TiO_2_) because smaller particles were more likely to enter the cell. The toxicological effects of ZnO nanoparticles are very important for cellulose materials, which are coming into direct contact with human skin. Therefore, this emphasizes the need for the intense investigation of ZnO nanoparticles and their effects on living organisms, including microorganisms [27].

Cellulose materials can be functionalized by applying layers of different nanoparticles to their structure. The functionalization of textiles with ZnO nanostructures enables the production of reinforced textiles resistant to tear and make wearing antibacterial textiles, UV- blocking textiles and other interesting materials possible. In addition to antibacterial properties, anti-odor, self-cleaning and medicinal textiles are made with ZnO. Functionalization can be achieved through different modifications, and the sol–gel process is an important part of them by being a simple, practical and cost-effective approach [28]. Moreover, it enables facile coating by obtaining a homogenous layer on large sample surfaces, the easy control of reaction kinetics, lower sintering temperatures and other advantages [28,29,30]. The most important advantages of the sol–gel process is the fact that a homogenous, unified and flat coating can be achieved at a mild and low temperature (below 100 °C), its flexibility and large surface modification [31]. 

## 2. Experimental 

### 2.1. Samples

Samples used for modification were made of viscose and cellulose polymers (ISO 1833: 2003), with a mass per unit area of 72.26 g/m^2^ (EN ISO 2286-2:2008), 0.51 mm thickness (EN ISO 5084: 2003) and with 54 (50) threads/cm in warp (weft), respectively (EN 1049-2: 2003). All used chemicals were of analytical grade and were utilized without additional purification. The glassware were properly washed, sanitized and autoclaved. ZnO nanoparticles used for modification of cellulose materials were purchased from Sigma Aldrich (product number 721077, 20 wt % dispersion of ZnO nanoparticles in water with certified particle size less than 100 nm), and Pt nanoparticles of 3 nm dispersion in water, product number 773875.

### 2.2. Dip Coating Process 

Cellulose materials were modified with ZnO nanoparticles by dip-coating methodology using 3-glycidyloxypropyltrimethoxysilane (GLYMO, Sigma Aldrich, Darmstadt, Germany, Europe) as a precursor. The speed of the process was 1 mm/s. The sols with ZnO nanoparticles were stirred magnetically and, afterwards, the GLYMO was added, and the process was carried out by ultrasound with different irradiation levels, and at different time intervals, until a homogeneous solution was obtained. Samples were left to gel at room temperature and then dried for 24 h at room temperature and atmospheric pressure, then at 100 °C for 60 min. Cellulose-modified products were characterized by thin layer chromatography (TLC), scanning electron microscopy with an EDX detector (SEM-EDX, Tescan Vega, Brno, Czech Republic) and FTIR-ATR (Perkin Elmer, Waltham, MA, USA) spectroscopy. The sol–gel process, influenced by numerous parameters such as temperature, time, precursor and HCl for catalysis, was optimized for all relevant parameters. 

In this research, the effect of ultrasonic irradiation power, the concentration of used reagents including nanoparticles, organic precursor and acid, as well as time intervals, on the structure, morphology and antimicrobial activity of cellulose materials modified with zinc oxide nanoparticles using the sol–gel method was investigated.

### 2.3. Spectroscopical and Microscopical Characterization

The structure and the morphology of samples before and after the modification with the ZnO nanoparticles were recorded under the scanning electronic microscope “TESCAN VEGA TS5136LS” (Tescan Vega company, Brno, Czech Republic) with an EDS detector, which was applied for this characterization of modified cellulose. An SEM microphotograph of the modified antimicrobial cellulose material is presented in Figure 1.

A Fourier transform infrared spectrometer (Spectrum 100 FTIR, Perkin Elmer, Waltham, MA, USA), which applies KBr and Attenuated Total Reflectance (ATR) techniques and enables the recording of spectra of cellulose materials in their solid state, was used to detect the functional groups before and after the modification. The FTIR spectra of samples were recorded at a resolution of 4cm^−1^ in the frequency range from 400 to 4000 cm^−1^ in diffuse reflectance mode. The identification of various functional groups and chemical structures in the ZnO-NPs was done by analyzing the absorption of electromagnetic waves at distinctive frequencies and intensities. An average of three scans for each sample was taken for the peak identification. A Nanoparticle Tracking Analyzer (NTA), “Nanosight”, was used to record the size and distribution of nanoparticles.

### 2.4. Testing of Antimicrobial Activity of Coating with Nanoparticles

The samples were tested for bactericidal and fungicidal activity by the disc-diffusion method, as well as growth inhibition studies against both Gram-positive (*Staphylococcus aureus*) and Gram-negative (*Escherichia coli)* bacteria. In order to assess the effects of ultrasonic irradiation on the antimicrobial effects of modified polymers, the *in vitro* antimicrobial activity of used colloidal ZnO nanoparticles for different model microorganisms were investigated by agar well diffusion and serial two-fold microdilution assays. After this preliminary step, cellulose materials modified with ZnO nanoparticles were tested for their antimicrobial properties with the “time-kill” assay. Performed experiments included standard laboratory Gram-positive and Gram-negative bacterial strains including American Type Culture Collection (ATCC) *Staphylococcus aureus* ATCC 29213 and *Escherichia coli* ATCC 10536, with additional methicillin-susceptible *S. aureus* (MSSA MFBF 10663) and methicillin-resistant *S. aureus* (MRSA MFBF 10679) clinical isolates from the stock-cultures of the Collection of Microorganisms of the Department of Microbiology, Faculty of Pharmacy and Biochemistry, University of Zagreb. All microbial media were purchased from Merck (Darmstadt, Germany) and norofloxacin and gentamicin were purchased from Sigma-Aldrich.

#### 2.4.1. Agar Well Diffusion Assay

The agar well diffusion assay was performed according to European Pharmacopoeia, with slight modifications [32]. Inocula were prepared from fresh overnight cultures with physiological saline and adjusted to 0.5 McFarland units (Kisker densitometer, Steinfurt, Germany). After inoculation, sample application to wells (50 µL) and preincubation of plates at +4 °C for 1 h followed by incubation at +37 °C for 18 h was performed under aerobic conditions in the dark. After incubation, antimicrobial activity was evaluated by measuring the diameters of zones of growth inhibition (d, mm) around wells. Gentamicin sulphate and norfloxacin were used for quality control of the method and strain susceptibility. All tests were performed in quintuplicate and results were expressed as the mean.

#### 2.4.2. Serial Microdilution Broth Assay

Minimal inhibitory concentrations (MICs) were investigated, guided by the serial microdilution broth assay, according to The European Committee on Antimicrobial Susceptibility Testing guidelines (EUCAST E.Def.5.1), with a major modification by replacing Mueller–Hinton broth with physiological saline in order to avoid the interfering activity of nanosized ZnO with the diverse protein contents present in the broth [33]. After performing the standard procedure of serial two-fold micro-dilution and incubation at 37 °C for 18–24 h, the colloidal nature of the nanosized ZnO and physiological saline as the medium conditioned the subcultivation of 10 µL from each dilution of the sample on the surface of tryptic soy agar. Reincubation at 35−37 °C was then carried out for 18 h in order to evaluate bacterial viability. Gentamicin sulphate and norfloxacin were used as the positive control, and pure physiological saline was used as the negative control. Minimal inhibitory concentration (MIC) was defined as the lowest concentration of ZnO, which allows no more than 20% microbial growth in comparison with the negative control. All tests were performed in triplicate and results were expressed as mean values.

#### 2.4.3. “Time-kill” Assay

Time-kill methodology was used to investigate the antimicrobial efficiency of the treated materials on MSSA and MRSA. The time-kill assay is a microbiological method that enables the quantification of microbial growth and survival under the influence of a specific sample monitored in time [34]. In the time-kill methodology, treated samples and the negative control (untreated samples) were cut into standard sized 1 × 1 cm pieces, inoculated with 100 μL of inoculum, incubated at 37 °C in an orbital shaker and examined at different time intervals (initial contact time and after 1, 3, 6, 12, 18 and 24 h). After every investigated time point, the sample was transferred in a sterile test tube, which was pre-filled with 1 mL of physiological saline and mixed for exactly 30 s (Vortex Genius 3, Ika, Germany). In total, 100 μL of the obtained solution was transferred in a new sterile tube, diluted 10 times, and a further 10-fold serial dilution was performed. Aliquots of diluted series of examined sample solutions were spread on tryptic soy agar plates to allow for the counting of bacterial colonies after 24 h incubation (aerobically at 37 °C, in dark). Colonies of viable bacteria were counted for both bacterial strains treated with the examined sample, as well as for the negative control (untreated cellulose fiber), and the results are shown graphically, with log10 CFU/mL (colony forming units per ml) as a function of time (h).

### 2.5. Response Surface Methodology Optimization 

In this study, the response surface methodology (RSM) based on Design of Experiment (State Ease version 9.1, State Ease Company, Minneapolis, MN, USA) software was applied to optimize the process of cellulose modification [35,36]. D-optimal design was selected for obtaining optimal results with a minimal number of preliminary experiments [37,38]. After the selection of the design, a “candidate set” with selected points was generated.

The selection procedure for the experimental setup reflected the preliminary experiments involving computer-intensive matrix modeling to achieve the highest possible precision of the model. In total, 26 preliminary experiments were designed and conducted, as is shown in Table 1. Initial parameters chosen for modeling were time, temperature, ultrasound irradiation power and concentrations of reagents (GLYMO, HCl and ZnO nanoparticles).

### 2.6. Tensile Properties

The elastic behavior of modified polymers was tested according to the norm, HRN EN ISO 22313: 2008, in which the material is exposed to the bending load at a 180° angle curve with the load on the press, released, and then the time needed to achieve the sample’s original position is measured.

### 2.7. Thin-Layer Chromatography 

For preliminary testing, in order to see which metals are present in the sample, thin-layer chromatography was applied. The standard sample solution was spotted on 20 × 20 cm TLC cellulose pre-coated plates (Merck, Darmstadt, Germany) by glass capillary. Development was, after the saturation of the chromatographic chamber, carried out in the Camag chromatographic chamber by the ascending technique to 8 cm distance. For the mobile system, acetonitrile–hydrochloric acid–water (72: 25: 23) was chosen. After development, the plates were dried, and exposed to ammonia vapors. The chromatograms with colored spots occurred after 37 min of the development, and showed the presence of zinc. R_F_ factors for each metal was calculated by using the following calculation: R_F_ = l_i_/l_0_(1)
presenting the ration between the distances reached by the sample and the mobile phase. After testing the standard metal solutions, the solutions of samples were investigated in the same way.

## 3. Results and Discussion

Cellulose materials with coatings of metal oxide nanoparticles are excellent antimicrobial agents. In the preparation of new functionalized polymers, sol–gel modification is widely used. This process includes many different parameters, which influence applicable properties, so, in this work, the efficiency of sol–gel modification was enhanced with the concentrated HCl, and a homogenous distribution of nanoparticles in the coating was achieved by ultrasonic homogenization. The surface coating was obtained by precursor 3-glycidyloxypropyltrimethoxysilane (GLYMO), which was characterized by FTIR spectroscopy and a scanning electron microscope equipped with an EDX detector (SEM-EDX). This coating was filled with ZnO nanoparticles with certified particle sizes less than 100nm, which were chemically bonded to the cellulose surface in order to obtain antibacterial functionalized cellulose material. The effect of ultrasonic irradiation power (40 to 90 kHz), the concentration of reagents (nanoparticles, precursor and acids) and time (15 to 72 min) were investigated in relation to the structure, morphology and antimicrobial activity of coatings with zinc oxide nanoparticles. The antibacterial effects of colloidal ZnO nanoparticles were tested against two reference bacterial reference strains *(S. aureus* and E. coli) and the killing effect of modified cellulose materials with 100 nm ZnO nanoparticles was tested against a methicillin-sensitive and a methicillin-resistant strain of *S. aureus.*

### 3.1. Spectroscopical, Chromatographical and Microscopical Investigations 

In this investigation, both SEM-EDX and FTIR spectroscopic methods were used to proof the quality of achieved modification of cellulose materials before the antimicrobial testing. SEM-EDX methodology enabled the fast and simple non-destructive investigation of the sample morphology and surface chemical composition [39,40,41]. The samples were observed under 80× magnification, after a process in which the samples were coated with the Au/Pd uniform coating (Figure 1). The size of the colloidally stable ZnO nanoparticles used in the modification of cellulose materials for antimicrobial testing was investigated and monitored by the Nanoparticle Tracking Analyzer Nanosight, which can determine the concentration of nanoparticles, as well as their distribution in the system. All nanoparticles were 100–120 nm in size.

Zinc monitored by thin-layer chromatography was determined under the R_F_ value of 0.16, and could be visualized by the pink color when detected with Alisarin S reagent.

Secondly, the samples were investigated by Fourier-transform infrared spectrometry (Spectrum 100 FTIR, Perkin Elmer, Waltham, MA), which applies KBr and Attenuated Total Reflectance (ATR) techniques and enables the recording of the spectra of cellulose materials in their solid state (Figure 2). FTIR was applied as an excellent technique for the investigation of different solid samples [40], including cellulose samples before and after the modification with nanoparticles.

As can be seen from Figure 2, the lines of samples before and after the sol–gel modification had distinguished differences: the epoxy groups were determined around ~905 and 911 cm^−1^, the area around 1100 cm^−1^ is linked to Si–O groups, (particularly Si–O–C and Si–O–Si bridges), and the conversion of metoxy groups in precursor GLYMO are distinguished at ~2870 cm^−1^. Epoxy groups of GLYMO only react with primary amine and silanol.

The processes of the hydrolysis and crosslinking were monitored and optimized with the goal being to determine favorable conditions for the preparation of the most efficient antimicrobial coating. To improve the application properties, the morphology of the inorganic phase within the hybrid was controlled by changing the hydrolysis conditions of GLYMO and other parameters. It is well known that hybrids from hydrolyzed GLYMO show better mechanical properties in acid hydrolysis, resulting in a very homogeneous hybrid structure, which is the reason why HCl was used in this research. Moreover, it is emphasized that the presence of hydrochloric acid serves as a catalyst for hydrolysis and decreases the thermal stability of those hybrids. GLYMO reacts in reactions of hydrolysis and condensation, in which alkoxide groups and epoxy rings open. Therefore, it can be concluded that this material is a good starting point for creating hybrid materials with covalent bonds between the organic and inorganic phases.

Moreover, the SEM-EDX results proved that cellulose materials were free of impurities and thus that they were successfully modified. Figure 3 shows an SEM microphotograph of the surface with nanoparticle modification and Figure 3b shows the EDX spectra of this surface. The mapping of the ZnO nanoparticles on a selected area is shown in Figure 3c. As can be seen from these results, a uniform homogenous coating was created over the sample surface with an equal distribution of ZnO nanoparticles (Figure 3c).

### 3.2. Optimization of Dip-Coating Process through Modeling of Recovery Angle

For the evaluation of the experimentally obtained results, a mathematical model was obtained. Its response is a function of the tested parameters and, in this investigation, the independent variables were the concentrations of reagents (GLYMO, HCl and ZnO nanoparticles), ultrasound irradiation power and time. 

Interactions between the independent variables are presented in Figure 4. This figure shows 3D response surface plots for the investigated parameters. The effects of the ultrasonound power are related to a) the concentration of GLYMO, b) the concentration of HCl, c) the concentration of ZnO nanoparticles and d) the time of the obtained modification. 

The 3D response surface condition plots were drawn by State Ease software and they enable the detection of the correlation between particular independent variables.

Figure 4 shows that at higher concentrations of precursor GLYMO and lower concentrations of HCl acid, the best results are achieved. Accordingly, at lower concentrations of ZnO and higher ultrasound power in shorter time intervals, higher numbers of nanoparticles were loaded onto the materials. Conversely, a prolonged sonication time and stronger power leads to the less effective loading of nanoparticles on the cellulose material.

This can be interpreted with a hypothesis that cellulose material has a limited surface available for nanoparticles [41]. Aggregated nanoparticles can be easily removed from the cellulose surfaces by bubble collapse during sonication. Moreover, small particles can penetrate into the material and adhere strongly to it, so using more nanoparticles results in more aggregation through prolonged sonication and results in removing aggregates with nanoparticles. In addition, the loading of nanoparticles relies upon both nanoparticle concentration and sonication time [41].

A good relaxation phenomenon is characterized with high values for the recovery angle. Therefore, this is a crucial parameter for medical textiles since it defines the comfort of the material and its softness in the contact with the skin. Since the modification of the surface of cellulosic materials was performed with the goal of making materials that could be used as medicinal textiles, the maximization of the recovery angle was selected as the goal of optimization. Cellulosic materials that were modified in 26 preliminary experiments were tested for their recovery angles according to the norm EN ISO 22313: 2008. Table 1 shows results that are recalculated as mean values from *n* = 3 experiments.

The results were used for modeling. The obtained model, describing the process with a good level of precision, is as follows:“Recovery angle” = 108.92 + 0.34c(GLYMO(mol/dm^−3^)) − 0.018c(HCl(mol/dm^−3^)) + 72.28(m/g(ZnO)) + 0.19(Ultrasound power/Hz) − 0.03 (time/min)(2)

To investigate if this model can be used in predicting the response for the investigated parameters and in the optimization of the process of cellulose modification, a comparison of predicted and experimentally obtained values was performed. The results are presented in Table 2.

The optimal parameter values for the modification of cellulose were predicted as 50 mL of GLYMO, m(ZnO) = 0.071 g, 50 mL of the catalyzer, HCl = 0.1 mol/L, and ultrasound 80 kHZ applied for 30.5 min. The predicted values of the parameters were experimentally tested, and the results were compared to predicted values (Table 2). As can be seen from the results, the obtained relative error between calculated and predicted values was very low (0.007–0.670%). Low values of the coefficient of variation (CV%) of the selected model indicate a very good model precision and the very good reliability of the experiments. Therefore, the optimal combination of parameters was used for the preparation of the antibacterial coating prior to the testing of antibacterial strength.

### 3.3. Antibacterial Strength of the Cellulose Modified with ZnO Nanoparticles

The optimal concentration of ZnO nanoparticles was prepared and zones of inhibition of bacterial growth and MICs were investigated for *E. coli* and *S. aureus*. The results shown in Table 3 reveal stronger antibacterial activity against *S. aureus* compared to *E. coli* in both performed assays (Z.I. 37 mm vs. 24 mm and MIC 0.03 ppm vs. MIC 0.76 ppm, respectively) [42]. 

In addition, ZnO nanoparticles were tested on two strains of *S. aureus*—MSSA and MRSA. The low MIC value of 3.28 ± 0.55 ppm for MRSA, alongside the even lower value for the *S. aureus* ATCC strain (0.03 ± 0.00 ppm), confirmed the potential of ZnO nanoparticles for further examination in antimicrobial coatings.

The coated cellulose materials were examined and the activity of samples with nanosized ZnO particles (100 nm) in combination with significantly smaller nanosized platinum particles (3 nm) are shown in Figure 5 and Figure 6.

As can be seen from Figure 5, the antibacterial effect starts significantly after 6h of exposure to cellulose materials modified with nanoparticles under ultrasonic irradiation compared to controls (untreated cellulose material). Figure 5 and Figure 6 show the results of antimicrobial experiments. All viable bacteria were eliminated after 24 h for all investigated samples. 

This is an excellent result, promising further valuable results in future research within this scientific area. The spread of MRSA can be prevented by the use of disposable gloves, capes and masks, but this is not always feasible. Therefore, new materials need to be developed that are active against microorganisms resistant to antibiotics. 

In this research, the effect of process parameters was investigated on the structure, morphology and antimicrobial activity of coatings with zinc oxide nanoparticles. The parameters were varied in their process ranges by ultrasonic irradiation power (40 to 90 kHz), the concentration of reagents (nanoparticles, precursor and acids) and time (15 to 72 min).

Based on the obtained results, it was observed that the amount of nanoparticles on cellulose material depended on the concentration of the reagents, ultrasound irradiation power and the time of the sonication. Other parameters had fewer effects. Moreover, ultrasonic irradiation (power and time) had a synergistic role in terms of the reagent concentration during the modification of the cellulose layer at low temperatures. 

## 4. Conclusions

In this research, ZnO nanoparticles were chemically bonded to the cellulose surface by the sol–gel process in order to obtain an antibacterial functionalized material. The ZnO nanoparticles were efficiently applied at low temperature and atmospheric pressure, and a homogenous coating was obtained. A prolonged sonication time and stronger power leads to the less effective loading of nanoparticles on the cellulose material, which confirms that ultrasonic irradiation is a vital parameter that is important for obtaining a homogenous, antimicrobial and effective surface.

SEM images and FTIR spectra proved the formation of new chemical bonds and the adhesion of ZnO nanoparticles on the surface of the material. Antibacterial testing proved that this material is an effective antibacterial surface.

Based on the statistical analysis, the amount of the nanoparticles on cellulose material depends on the concentration of the reagents, ultrasound irradiation power and the time of the sonication. According to the obtained model, the optimal conditions for achieving the most flexible antimicrobial material depend on the concentration of reagents (ZnO 0.071 g, HCl 0.1 M) and ultrasound irradiation (80 kHz). For these conditions, a very low relative error between predicted and experimental values was obtained (0.007–0.670%). Therefore, it can be concluded that ultrasonic irradiation (power and time) has a synergistic role in terms of the reagent concentration during the modification of cellulose layers at low temperatures. 

## Figures and Tables

**Figure 1 polymers-12-01210-f001:**
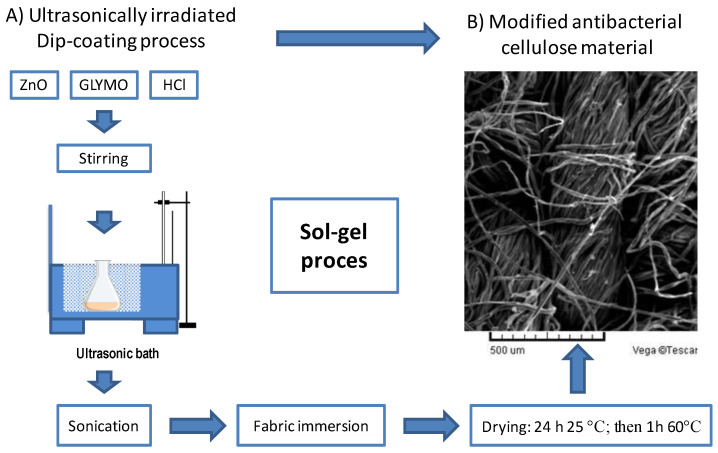
Schematic overview of ultrasonically irradiated dip-coating process with (**A**) diagram of sonicated sol–gel process and (**B**) the resulting SEM microphotograph recorded under a magnification of 80× on the modified antimicrobial cellulose material.

**Figure 2 polymers-12-01210-f002:**
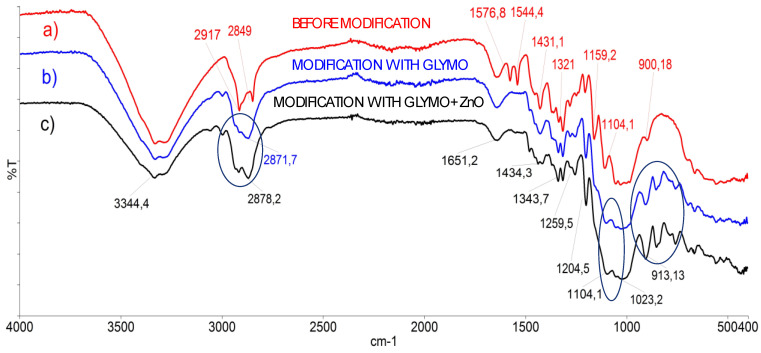
Fourier-transform infrared spectra of cellulose sample before and after the modification with the ZnO nanoparticles: (**a**) Red line: FTIR-Attenuated Total Reflectance (ATR) spectra of cellulose sample before modification, (**b**) Blue line: FTIR ATR spectra of cellulose sample after modification with 3-glycidyloxypropyltrimethoxysilane (GLYMO), (**c**) black line: FTIR ATR spectra of cellulose sample modified with ZnO nanoparticles and GLYMO, with optimal modification parameters calculated by the Design of Experiment program. Three blue circles define area of the epoxy groups were determined around ~905 and 911 cm^−1^, the area around 1100 cm^−1^ is linked to Si–O groups, (particularly Si–O–C and Si–O–Si bridges), and the conversion of metoxy groups in precursor GLYMO are distinguished at ~2870 cm^−1^.

**Figure 3 polymers-12-01210-f003:**
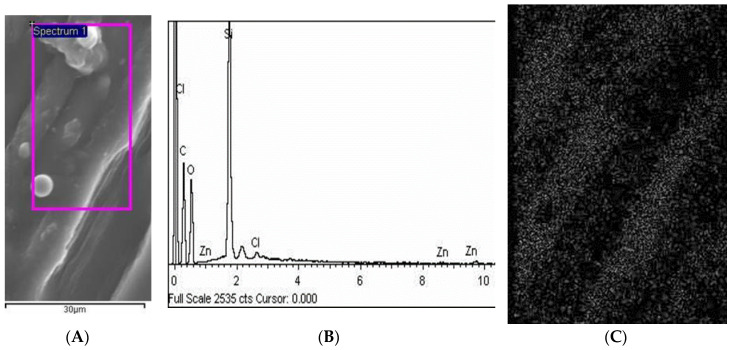
(**A**) SEM microphotograph of ZnO nanoparticle and homogenous coating layer on cellulose materials with specified area used for EDX investigation, The purple square is the area that was scanned for mapping (C); (**B**) Results of the SEM-EDX investigation with the proof that the cellulose material was successfully modified with the ZnO coating, (**C**) SEM-EDX mapping was additionally performed on a specified area of the cellulose material in order to detect ZnO nanoparticles (gray dots on the figure) showing a homogenous coating, as proof of successful modification. Magnification of C was 1000 times.

**Figure 4 polymers-12-01210-f004:**
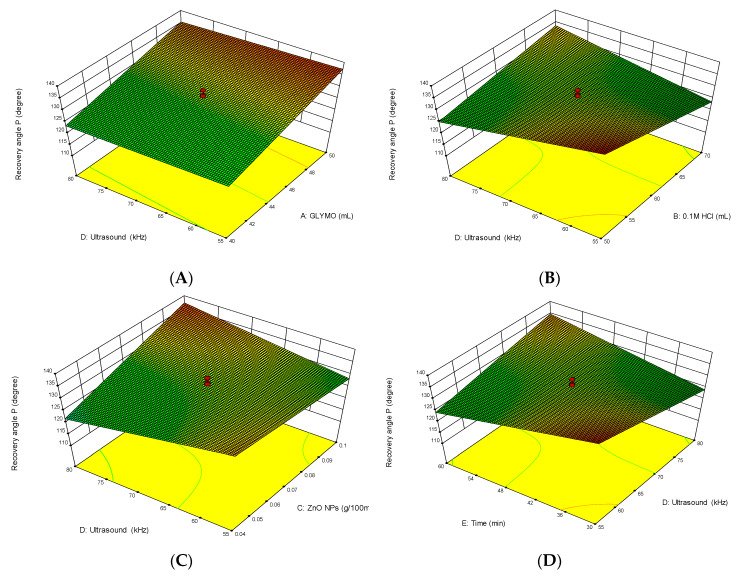
Three-dimensional response surface plots for investigating parameters: effects of the ultrasonic irradiation and (**A**) GLYMO, (**B**) HCl, (**C**) ZnO nanoparticles and (**D**) time on the obtained modified cellulose.

**Figure 5 polymers-12-01210-f005:**
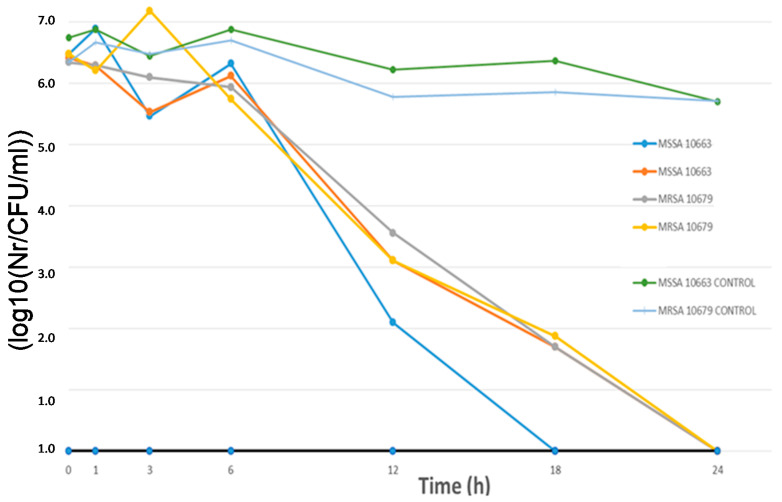
“Time-kill” assay—graphical presentation of antimicrobial efficiency of modified cellulose materials with 100 nm ZnO nanoparticles. The effects were achieved in combination with colloidal platinum (3 nm) against MSSA and MRSA.

**Figure 6 polymers-12-01210-f006:**
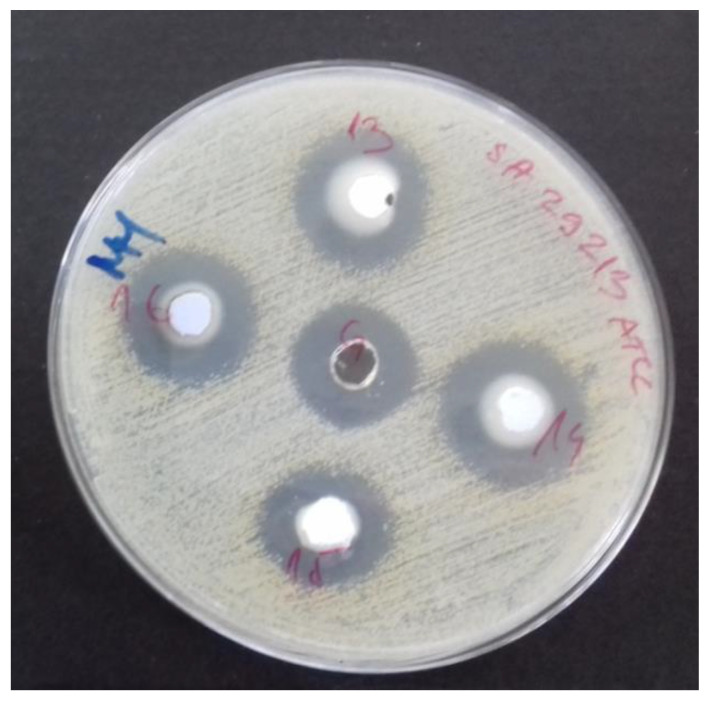
The plate after antimicrobial tests show potent antimicrobial activity against tested microorganisms.

**Table 1 polymers-12-01210-t001:** Investigated parameters used during optimization of dip-coating process by Design of Experiment statistical methodology and calculated recovery angles.

Nr	GLYMO, mL	0.1 M HCl, mL	ZnO, g/100 mL	Ultrasound, Hz	Time, min	RECOVERY ANGLES
**1**	45.0	41.8	0.70	67.5	45.0	132.6
**2**	45.0	78.2	0.70	67.5	45.0	126.6
**3**	45.0	60.0	0.70	67.5	45.0	117.7
**4**	50.0	70.0	0.40	55.0	60.0	121.7
**5**	35.9	60.0	0.70	67.5	45.0	112.8
**6**	40.0	70.0	1.00	55.0	60.0	130.7
**7**	45.0	60.0	0.70	67.5	17.7	128.8
**8**	40.0	70.0	0.40	80.0	60.0	129.3
**9**	45.0	60.0	0.70	44.7	45.0	136.2
**10**	45.0	60.0	0.15	67.5	45.0	129.2
**11**	45.0	60.0	0.70	90.3	45.0	133.2
**12**	45.0	60.0	0.70	67.5	45.0	128.2
**13**	50.0	70.0	1.00	55.0	30.0	125.5
**14**	45.0	60.0	0.70	67.5	45.0	133.5
**15**	40.0	50.0	0.40	55.0	30.0	130.1
**16**	50.0	50.0	1.00	55.0	60.0	130.6
**17**	45.0	60.0	0.70	67.5	45.0	131.6
**18**	45.0	60.0	0.70	67.5	72.3	126.1
**19**	45.0	60.0	0.70	67.5	45.0	131.3
**20**	50.0	50.0	0.40	80.0	60.0	137.5
**21**	40.0	70.0	1.00	80.0	30.0	138.8
**22**	54.1	60.0	0.70	67.5	45.0	134.6
**23**	45.0	60.0	1.25	67.5	45.0	138.2
**24**	40.0	50.0	1.00	80.0	60.0	138.8
**25**	50.0	50.0	1.00	80.0	30.0	137.5
**26**	50.0	70.0	0.40	80.0	30.0	133.0

**Table 2 polymers-12-01210-t002:** Comparison of predicted and actual values of recovery angles.

GLYMO, mL	0.1M HCl, mL	ZnO NPs g/100 mL	Ultrasound, Hz	Time, min	Predicted Recovery Angle	Actual Recovery Angle
50.000	50.000	0.071	79.975	31.340	133.6	132.7
50.000	50.297	0.071	80.000	30.572	135.7	135.8

**Table 3 polymers-12-01210-t003:** Antibacterial activity of colloidal ZnO nanoparticles.

Agar Well Diffusion (Z.I., mm)					Microdilution (MIC)		
Bacterial Strain	ZnO		Positive Control (μg/mL) **		ZnO μg/mL	Positive Control (μg/mL)	
	21% *	10% *	Norfloxacin	Gentamicin		Norfloxacin	Gentamicin
*S. aureus* ATCC 29213	37	15	-	17	0.03	-	0.63
*E. coli* ATCC 10536	24	0	23	-	0.76	0.02	-

* 21% is equal to 210 000 μg/mL, 10% equal to 100 000 μg/mL; ** Norfloxacin and gentamicin stock concentrations = 10 µg/mL; “-” not performed.

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
