# Peer review of "Synthesis, Modification and Characterization of Antimicrobial Textile Surface Containing ZnO Nanoparticles"

_polymers, 2020, doi:10.3390/polym12061210_

Round 1
Reviewer 1 Report
Usage of protective and medical antibacterial textile is currently limited due
Usage of protective and medical antibacterial textile is currently limited due to the high cost of such materials. Therefore, innovative technologies for production of antibacterial materials are investigated.
In this research, ZnO nanoparticles were chemically bonded to the cellulose surface by sol-gel process, in order to obtain antibacterial functionalized material. The effect of ultrasonic irradiation power (40 to 90 kHz), concentration of reagents (nanoparticles, precursor and acids) and time (15 to 72 min) was investigated on the structure, morphology and antimicrobial activity of coatings with zinc oxide nanoparticle. The antibacterial effects of colloidal ZnO nanoparticles were tested against two reference bacterial reference strains (S. aureus and E. coli) and the killing effect of modified cellulose materials with 100 nm ZnO nanoparticles was tested against a methicillin-sensitive and a methicillin-resistant strain of S. aureus.
The Authors observed that the amount of the nanoparticles on cellulose material depends on concentration of the reagents, ultrasound irrigation power and the time of the sonication. Particularly, ultrasonic irradiation (power and time) has synergistic role with the reagents concentration during modification of cellulose layer at low temperatures.
Comments:
The results of this study are interesting and they may be translated in therapeutic application. The Results/Discussion section requires a thorough review since they have several gaps. Both Results and Discussion should be more detailed.
Introduction: Lines 44-46. Clarify what means the “core” of bacteria, it is unusual definition.
Figure 5. In figure 5 were shown unicod 11 and 12 for every bacterial strain tested. What means? What are unicod 11 and unicod 12 strains? They are not cited in any section of the text.
Figure 6. What represents this figure? The text and the figure legend don’t explain what the figure shows.
In the par. 2.5 “Testing of antimicrobial activity of coating with nanoparticles” in materials and methods was cited Candida albicans, but in the results was not cited any experiment with this microorganism.
Author Response
Response to Reviewer 1 Comments
Point 1: The results of this study are interesting and they may be translated in therapeutic application. The Results/Discussion section requires a thorough review since they have several gaps. Both Results and Discussion should be more detailed.
Response 1: Dear Reviewer, thank you very much for your valuable comments.
More details were added to the manuscript into the Results/Discussion section as required by Reviewer's 1 Comments. Newly added text is marked in red color, as is shown in pages 8, 9, 14, 15, 16 and 17. We believe that the text of the manuscript is now much more suitable for publication.
Point 2: Introduction: Lines 44-46. Clarify what means the “core” of bacteria, it is unusual definition.
Response 2: The sentence is now more clearly specified as: “bacterial surface and/or with the bacterial cell, after it enters through the membrane”
Point 3: Figure 5. In figure 5 were shown unicod 11 and 12 for every bacterial strain tested. What means? What are unicod 11 and unicod 12 strains? They are not cited in any section of the text.
Response 3: Unicod which was a code from the laboratory experiments is removed from the figure 5.
Point 4: Figure 6. What represents this figure? The text and the figure legend don’t explain what the figure shows.
Response 4: Figure 6 is now explained in the texts of the manuscript (page 17, line 528: “As is presented in Figure 6, ZnO nanoparticles showed efficient antibacterial resistance not only against S. aureus (left picture), but also against E. colli microorganisms. Moreover, the coatings on cellulose materials containing ZnO nanoparticles also proved antibacterial effects against reference bacterial reference strains (S. aureus and E. coli) and the killing effect of modified cellulose materials with 100 nm ZnO nanoparticles was tested against a methicillin-sensitive and a methicillin-resistant strain of S. aureus. Based on the recived results, it was observed that the amount of the nanoparticles on cellulose material depended on concentration of the reagents, ultrasound irrigation power and the time of the sonication. Other parameters had fewer effects. Moreover, ultrasonic irradiation (power and time) had synergistic role with the reagents concentration during modification of cellulose layer at low temperatures.”)
Point 5: In the par. 2.5 “Testing of antimicrobial activity of coating with nanoparticles” in materials and methods was cited Candida albicans, but in the results was not cited any experiment with this microorganism.
Response 5: Although our preliminary experiments covered also research against Candida albicans, those results are not presented in this work. Therefore this part of the sentence is removed from the manuscript.

Reviewer 2 Report
The manuscript reports on antimicrobial activity of textiles impregnated with ZnO nanoparticles. The report is very well written with only a few minor typological errors.
The research is designed properly, the results are presented clearly and aesthetically. The discussion is clear and properly concluded.
There is only one minor issue which has to be addressed prior the final acceptance, namely, the origin of ZnO nanoparticles has to be clarified. In more detail, the authors should specify whether they synthesised the ZnO nanoparticles on their own or whether they purchased the commercially Available product. If synthesised, please provide the detailed procedure and if purchased, please add a relevant comment in the materials and methods section.
Overall, the manuscript is interesting and original, and meets the criteria for being published in Polymers.
Therefore I recommend acceptance after minor revision.
Author Response
Response to Reviewer 2 Comments
Dear Reviewer 2, thank you very much for your valuable comments. Here is the response to the points raised:
Point 1: There is only one minor issue which has to be addressed prior the final acceptance, namely, the origin of ZnO nanoparticles has to be clarified. In more detail, the authors should specify whether they synthesised the ZnO nanoparticles on their own or whether they purchased the commercially Available product. If synthesised, please provide the detailed procedure and if purchased, please add a relevant comment in the materials and methods section.
Response 1: Commercially available ZnO nanoparticles were used in this research group, which is now clearly specified in the materials and methods section:
Page 4, Lines: 134 – 137:
“ZnO nanoparticles used for modification of cellulose materials were purchased from Sigma Aldrich (product Number 721077, 20 wt% dispersion of ZnO nanoparticles in water with certified particle size less than 100 nm).“

Round 2
Reviewer 1 Report
I think that further changes are needed:
- In the text and in the legend of figure 5 Authors should state the two concentrations of ZnO nanoparticles used.
- figure 6 should be more clear if the Authors indicate the bacterial species showed in the picture.
Kindly regards,
Author Response
Response to Reviewer 2 Comments
Response to Reviewer 2 Comments
Dear Reviewer 2, thank you very much for your valuable comments. Here is the response to the points raised:
Point 1: I think that further changes are needed: “In the text and in the legend of figure 5 Authors should state the two concentrations of ZnO nanoparticles used.”
Response 1: It is not two concentrations of ZnO nanoparticles, but only one, ZnO of 100 nm. The other nanoparticle that was added to results is the mixture with Pt nanoparticle of 3 nm in size, as is described before and after the Figure. To clarify this in larger extant, we have emphasized this fact in page 4, line 134 – 137, and in page 14, lines 431 – 433. Therefore, it is now clearly presented in the text and in the legend of the Figure 5.
Point 2: “Figure 6 should be more clear if the Authors indicate the bacterial species showed in the picture.”
Response 2: Figure 6 is now clarified with more detailed description: “Figure 6. The plates after antimicrobial tests by diffusion and dilution methods show potent antimicrobial activity against S. aureus (left picture shows the results of the diffusion methodology), as well as against E. colli microorganisms (right part of the Figure shows the micro dilution methodology)”.
